# Learning Video Representations using Contrastive Bidirectional Transformer

## Abstract

This paper proposes a self-supervised learning approach for video features that results in significantly improved performance on downstream tasks (such as video classification, captioning and segmentation) compared to existing methods. Our method extends the BERT model for text sequences to the case of sequences of real-valued feature vectors, by replacing the softmax loss with noise contrastive estimation (NCE). We also show how to learn representations from sequences of visual features and sequences of words derived from ASR (automatic speech recognition), and show that such cross-modal training (when possible) helps even more.

## 1 Introduction

Recently there has been a lot of progress in self-supervised representation learning for textual sequences, followed by supervised fine-tuning (using small labeled datasets) of shallow (often linear) decoders on various downstream NLP tasks, such as sentiment classification. In this paper, we build on this work and propose a new method for self-supervised representation learning for videos, optionally accompanied by speech transcripts generated by automatic speech recognition (ASR). We show that fine-tuning linear decoders together with our self-supervised video representations, can achieve state of the art results on various supervised tasks, including video classification, segmentation and captioning.

Our approach builds on the popular BERT (Bidirectional Encoder Representations from Transformers) model (Devlin et al., 2018) for text. This uses the Transformer architecture (Vaswani et al., 2017) to encode long sentences, and trains the model using the "masked language modeling" (MLM) training objective, in which the model must predict the missing words given their bidirectional context. The MLM loss requires that each token in the sequence be discrete. The VideoBERT model of (Sun et al., 2019a) therefore applied vector quantization (VQ) to video frames before passing them (along with optional ASR tokens) to the BERT model. Unfortunately, VQ loses fine-grained information that is often critical for downstream tasks. More recently, several papers (e.g., VilBERT (Lu et al., 2019) and LXMERT (Tan & Bansal, 2019)) proposed to address this limitation by directly measuring the visual similarity between frames using pre-trained visual encoders.

In this paper, we propose a way to train bidirectional transformer models on sequences of real-valued vectors (e.g., video frames), $x_{1:T}$, using noise contrastive estimation (NCE), without needing pre-trained visual encoders. We call our method "Contastive Bidirectional Transformer" or CBT. We also develop a method that combines $x_{1:T}$ with an optional sequence of discrete tokens, $y_{1:T'}$ (e.g., derived from ASR). In contrast to the VideoBERT paper (Sun et al., 2019a), we provide a "lightweight" way of combining these signals after training each modality separately. In particular, we propose a cross-modal transformer to maximize the mutual information between $x_{1:T}$ and $y_{1:T'}$ at the sequence level (rather than at the frame level). This method is robust to small misalignments between the sequences (e.g., if the words at time $t$ do not exactly correspond to what is visually present in frame $t$).

We demonstrate the effectiveness of the proposed approach for learning short-term visual representations, as well as longer term temporal representations. For visual representations, we encode each window of $K$ frames using a 3D convolutional neural network S3D (Xie et al., 2018), and then pass this sequence of features to the CBT model for self-supervised pretraining with the NCE loss on the

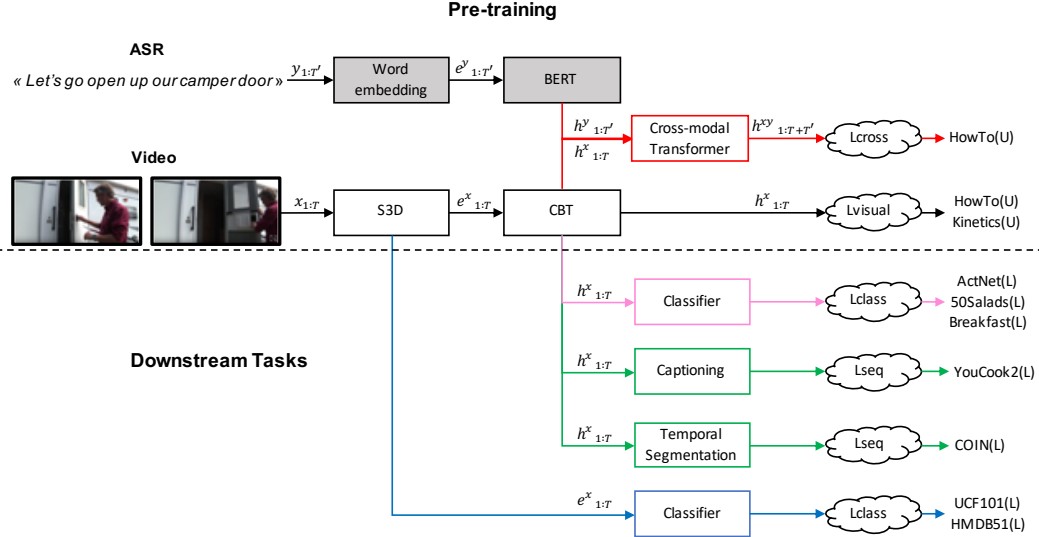

Figure 1: Summary of our method for training and evaluation. The blocks above the line are pre-trained in an self-supervised way. The solid blocks represent the BERT language model, which is pre-trained on web text and frozen (see section 3.1). The black CBT visual block is trained using NCE loss on unlabeled HowTo or Kinetics videos (see section 3.2). The red cross-modal transformer is trained using cross-modal loss on HowTo with ASR (see section 3.3). The components below the line are trained in a supervised way on various tasks. The purple block is trained for next action prediction on ActivityNet, Breakfast, and 50Salads (see section 4.2). The blue block is trained for video classification on UCF101 and HMDB501 (see section 4.1). The green blocks are trained on captioning and video segmentation tasks, which are described in the supplementary material (section 6.1 and section 6.2). Lseq refers to cross-entropy sequence loss.

unlabeled Kinetics dataset (Kay et al., 2017). We then fine-tune a linear classifier for video classification on UCF101 and HMDB51. We show that our method outperforms previous state-of-the-art self-supervised methods by large margins (UCF101 from 75.7% to 79.5% and HMDB51 from 35.7% to 44.6%).

For temporal representations, we encode each window of $K$ frames using a S3D network that is pretrained on Kinetics, and then "frozen". We then pass this sequence of features to the CBT model for self-supervised pretraining with the NCE loss on the unlabeled HowTo100M dataset (Miech et al., 2019b). We also evaluate the effects of running ASR on HowTo100M, and passing this to our cross-modal transformer as an additional signal. We then fine-tune various shallow decoders for a variety of tasks, including video classification, segmentation and captioning. We show large gains compared to previous methods, especially when we use cross-modal pretraining.

See fig. 1 for a summary of our training method and evaluation protocol.

## 2 RELATED WORK

**Video representations.** Most existing work on learning video representations, such as (Simonyan & Zisserman, 2014; Varol et al., 2018; Carreira & Zisserman, 2017; Xie et al., 2018; Tran et al., 2014), only captures a few seconds of video. Long-term context can be encoded by recurrent neural networks (Abu Farha et al., 2018; Sun et al., 2019b), graph convolutional networks (Zhang et al., 2019), or long-term feature banks (Wu et al., 2019), but these are all supervised methods. Some recent work have been done on learning self-supervised video representation (Vondrick et al., 2016; Wang & Gupta, 2015; Misra et al., 2016; Sermanet et al., 2018; Han et al., 2019; Xu et al., 2019; Wang et al., 2019a) by defining pretext tasks such as ordering (Lee et al., 2017), rotation (Jing et al., 2018), temporal cycle consistency (Wang et al., 2019b; Dwibedi et al., 2019) or colorization (Vondrick et al., 2018) but similar to their supervised counterparts they capture only few seconds.

**Self-supervised context modeling.** Recently, there has between a lot of work on self-supervised context modeling for language representations (Peters et al., 2018; Radford et al., 2019; Devlin et al., 2018). In particular, the BERT model, which stands for Bidirectional Encoder Representations from Transformers (Devlin et al., 2018), pre-trains deep bidirectional representations by jointly conditioning on both left and right context in all layers. The pre-trained BERT representations can be fine-tuned with just one additional output layer to create state-of-the-art models for a wide range of NLP tasks, such as question answering and linguistic entailment. Our representation builds on this approach and adapts it to continuous video data by using a contrastive loss.

**Mutual information estimation and maximization.** For representation learning, a signal encoder can be trained to maximize the mutual information (MI) between the input signal and its encoded outputs, or the encoded outputs of the signal and its context (see e.g., (Belghazi et al., 2018; Hjelm et al., 2019; Oord et al., 2018; Tian et al., 2019). In particular, contrastive predictive coding (CPC) (Oord et al., 2018) uses noise contrastive estimation (Gutmann & Hyvärinen, 2010) to maximize the MI lower bound. Unlike CPC, which relies on auto regressive models to encode context, we use BERT to encode bidirectional context within each sequence, and across different modalities.

**Cross-modal learning.** The multi-modality of video is a rich source of information for self-supervised learning of video representations. Since videos contain both visual and audio signals that are roughly synchronized, the two modalities can supervised each other, as explored in prior work such as (Aytar et al., 2016; Owens et al., 2016b;a; Zhao et al., 2018). Another common form of weak supervision is based on video and language, where language is either obtained by automatic speech recognition (ASR) or from additional textual description. Language can be leveraged by finding a joint embedding space for both visual and textual modalities or by learning an alignment between the two modalities (Miech et al., 2018; Alayrac et al., 2016; Zhou et al., 2018a).

Recently, several concurrent approaches (Sun et al., 2019a; Lu et al., 2019; Li et al., 2019a; Su et al., 2019; Tan & Bansal, 2019; Li et al., 2019b) generalize the BERT architecture and MLM objective to learn visual-linguistic representations. They assume the visual representations to be fixed and given by supervised pre-trained visual encoders, and define the visual MLM objective in terms of visual similarities (e.g. via vector quantization or measuring L2 distance) between the original and predicted visual representations. To the best of our knowledge, our proposed CBT is the first to demonstrate the effectiveness of BERT-style pre-training in a fully self-supervised way for video.

## 3 METHOD

We first give an overview of the BERT model for learning from sequences of words, $y_{1:T'}$. We then discuss an extension to the case of sequences of video frames, $x_{1:T}$. Finally, we discuss how to learn from both kinds of data, even when not perfectly aligned.

### 3.1 THE BERT MODEL

The BERT model (Devlin et al., 2018) takes in a sequence of discrete tokens, $y_{1:T'}$, where $y_t \in \{1, \ldots, K\}$, embeds each one into a dense vector, $e_t^y \in^D$, and then emits a sequence of dense output vectors, $h_t^y \in D^y$, which are computed using a transformer (Vaswani et al., 2017). The output sequence captures local and global semantic information about the input sequence.

The main training objective for BERT is to minimize the pseudo negative log likelihood, defined by

$$L_{\text{bert}} = -E_{\mathbf{y} \sim \mathcal{D}} \sum_{t=1}^{T} \log p(y_t | y_{-t}) \tag{1}$$

where $y_{-t}$ is the sequence of all words except the $t$'th, and

$$p(y_t | y_{-t}) = \frac{\exp(\mathbf{e}_t^T \hat{\mathbf{e}}_t)}{\sum_{k=1}^{K} \exp(f_{\text{enc}}(k)^T \hat{\mathbf{e}}_t)} \tag{2}$$

Here $f_{\text{enc}}(k)$ is an embedding lookup table for token $k$, $\mathbf{e}_t = f_{\text{enc}}(y_t)$ is the embedding for the token at $t$, $\mathbf{e}_{-t} = [f_{\text{enc}}(y_l) : l < t, 0, f_{\text{enc}}(y_l) : l > t]$ is the embedding sequence for the context, and

$\hat{e}_t = g_{\text{context}}(\mathbf{e}_{-t})$ is a multi-layer multi-headed transformer network that takes a $T \times D$ feature matrix as input (masked at location $t$) and returns a matrix of the same size.

## 3.2 THE CBT MODEL

The BERT model requires a fixed discrete vocabulary. However, for images and videos, the inputs are real-valued vectors. We propose to use the softmax version of the noise contrastive estimation (NCE) loss (Jozefowicz et al., 2016), which has the form

$$L_{\text{visual}} = -E_{\mathbf{x} \sim \mathcal{D}} \sum_t \log \text{NCE}(\mathbf{x}_t | \mathbf{x}_{-t}) \tag{3}$$

where

$$\text{NCE}(\mathbf{x}_t | \mathbf{x}_{-t}) = \frac{\exp(\mathbf{e}_t^T \hat{\mathbf{e}}_t)}{\exp(\mathbf{e}_t^T \hat{\mathbf{e}}_t) + \sum_{j \in \text{neg}(t)} \exp(\mathbf{e}_j^T \hat{\mathbf{e}}_t)} \tag{4}$$

where $\mathbf{e}_t = f_{\text{enc}}(\mathbf{x}_t)$ is the output of a 3D CNN applied to a small window around frame $t$ (we use the S3D model from (Xie et al., 2018)), $\hat{\mathbf{e}}_t = g_{\text{context}}(\mathbf{e}_{-t})$ is the output of a visual transformer, and $\text{neg}(t)$ is a set of (indices of) "negative examples" (in practice we use all the other frames from the same minibatch as frame $t$).

Intuitively the NCE loss encourages the model to learn to identify the correct frame (given the context) compared to a set of negative distractors. More formally, it can be shown that the NCE loss maximizes (a lower bound on) the mutual information (MI) between $\mathbf{x}_t$ and $\mathbf{x}_{-t}$ (see e.g., (Oord et al., 2018; Poole et al., 2019)). This loss has been used in other work on self-supervised visual representation learning, e.g., in the deep infomax (DIM) (Hjelm et al., 2019) and contrastive predictive coding (CPC) (Oord et al., 2018) papers. In DIM, the context predictor uses a CNN applied to neighboring patches in the same image, and in CPC, the context predictor uses a causal autoregressive model applied to "earlier" patches in the same image. In our CBT method, the context predictor is a bidirectional transformer applied to video frames.

## 3.3 THE CROSS-MODAL CBT MODEL

In this section we show how to learn useful representations from sequences of continuous visual features (from video) and sequences of discrete words (from ASR). More precisely, assume we have two sequences, $\mathbf{x} = \mathbf{x}_{1:T}$ representing video, and $\mathbf{y} = \mathbf{y}_{1:T'}$, representing ASR. Note that the sequences may not be perfectly aligned, since a person may speak about things at time $t$ that are not visible in the frame at time $t$. Therefore it does not make sense to try to maximize the MI between $\mathbf{x}_t$ and $\mathbf{y}_t$ at the frame level. Instead, we try to maximize MI between $\mathbf{x}$ and $\mathbf{y}$ at the sequence level.

To do this, we first encode each sequence using CBT and BERT to get $\mathbf{h}_{1:T}^x = \text{CBT}(\mathbf{x}_{1:T})$ and $\mathbf{h}_{1:T'}^y = \text{BERT}(\mathbf{y}_{1:T'})$, as shown in fig. 1. We then concatenate these sequences and pass them to a shallow cross-modal transformer to produce $\mathbf{h}_{1:T+T'}^{xy}$. Finally, we pass this to a shallow MLP to compute an MI-like score $\text{MI}(\mathbf{x}, \mathbf{y}) = f(\mathbf{h}_{1:T+T'}^{xy})$. (Here $f()$ extracts the features from $\mathbf{h}_0^{xy}$, but it could also use average pooling.) This model is trained using $L_{\text{cross}} = -E_{(\mathbf{x},\mathbf{y}) \sim \mathcal{D}} \log \text{NCE}(\mathbf{y}|\mathbf{x})$, where

$$\text{NCE}(\mathbf{y}|\mathbf{x}) = \frac{\text{MI}(\mathbf{x}, \mathbf{y})}{\text{MI}(\mathbf{x}, \mathbf{y}) + \sum_{\mathbf{y}' \in \text{Neg}(\mathbf{y})} \text{MI}(\mathbf{x}, \mathbf{y}')} \tag{5}$$

where $\text{Neg}(\mathbf{y})$ is a set of ASR sequences not associated with video $\mathbf{x}$.

Note that our cross-modal training assumes there is something in common between the video and text streams. In practice this means we have to focus on certain kinds of videos, such as instructional videos, in which the spoken words and the visual content are "about" the same thing. By contrast, arbitrary videos may contain speech content that is unrelated to the visual frames (e.g., imagine a conversation between two characters in a drama or soap opera).

## 3.4 OVERALL MODEL

Our overall model has three components: one transformer (BERT) that takes discrete ASR tokens, one transformer that takes continuous video features, and a third transformer to estimate mutual

information between the two modalities. We jointly train the model by optimizing:

$$L_{\text{cbt}} = w_{\text{bert}} L_{\text{bert}} + w_{\text{visual}} L_{\text{visual}} + w_{\text{cross}} L_{\text{cross}} \tag{6}$$

We fix $w_{\text{bert}} = 0$, since we use a pre-trained (frozen) BERT model for ASR. We set $w_{\text{visual}} = 1$, and either set $w_{\text{cross}} = 1$ or $w_{\text{cross}} = 0$, depending on whether we use cross-modal training or not.

## 4 EXPERIMENTS

In this section we conduct experiments to study the usefulness of the representations learned by our CBT model for various downstream tasks, including action anticipation, video captioning and action segmentation. We also consider ablations to our model, such as turning cross-modal training on or off, varying the size of the visual transformer, and varying the amount of unlabeled pre-training data.

### 4.1 LEARNING SELF-SUPERVISED VISUAL REPRESENTATIONS

In this section we evaluate self-supervised visual representation learning on the downstream task of action recognition. Existing methods use various proxy tasks to pre-train feature extractors in a self-supervised way, and then use supervised learning to train linear classifiers on top of these frozen representations, or fine-tune the classifier plus feature extractor end-to-end. We follow the same protocol.

**Experimental setup.** We follow the standard practice from recent works (Han et al., 2019; Jing et al., 2018; Wang et al., 2019a) by pre-training our model (S3D feature extractor followed by CBT) on unlabeled RGB-only Kinetics (Kay et al., 2017) videos. Kinetics is the largest action recognition dataset containing 500k short clip videos (about 10 seconds long) for 600 human actions classes. We take 1 minute sliding windows from the original YouTube videos they are selected from. We then use the (average pooled) S3D features as input to a linear classifier, and train the classifier on various datasets. For evaluation, we use UCF101 (Soomro et al., 2012), which contains 13,320 videos from 101 human actions, and HMDB51 (Kuehne et al., 2011), which contains 7,000 videos from 51 classes. For both dataset we report the action recognition test accuracy averaged over the 3 standard train/test splits.

To pre-train our CBT model, we use a curriculum learning strategy, by first pre-training the S3D feature extractor on unsupervised clips using the loss proposed in the 3DRotNet paper (Jing et al., 2018) on 16 consecutive frames. We then jointly fine-tune the last blocks of S3D (`Mixed5b` and `Mixed5c`) with the visual transformer using the CBT visual loss. We observed that this strategy gave us better results on downstream tasks compared to pre-training from scratch using CBT; it also saves memory and computation, which allows us to use longer sequences.

During pre-training, we set the number of visual transformer layers to be 2, number of attention heads to be 4, and hidden unit size to be 768. We randomly take 60-second sliding windows from the Kinetics videos, and break them into sequences of 1.5-second clips. We randomly mask out 6 out of the 40 possible locations. We resize the video frames to 112 by 112 before encoding them with S3D to save memory. The model is trained for 500K iterations with batch size of 128 and learning rate of 1e-5.

| Method | Fine-tuned | | Frozen | |
|---|---|---|---|---|
| | UCF101 | HMDB51 | UCF101 | HMDB51 |
| Random | 63.3 | 29.7 | 25.7 | 11.5 |
| Shuffle&Learn* | 68.7 | 35.8 | 26.5 | 12.6 |
| 3DRotNet* | 75.3 | 40.0 | 47.7 | 24.8 |
| CBT | **79.5** | **44.5** | **54.0** | **29.5** |

| Method | Dataset | UCF101 | HMDB51 |
|---|---|---|---|
| Shuffle&Learn (Misra et al., 2016) | UCF101 | 50.2 | 18.1 |
| OPN (Lee et al., 2017) | UCF101 | 59.6 | 23.8 |
| ClipOrder (Xu et al., 2019) | UCF101 | 72.4 | 30.9 |
| Wang et al. (2019a) | Kinetics | 61.2 | 33.4 |
| 3DRotNet (Jing et al., 2018) | Kinetics | 66.0 | 37.1 |
| DPC (Han et al., 2019) | Kinetics | 75.7 | 35.7 |
| CBT | Kinetics | **79.5** | **44.6** |

Table 1: Self-supervised action recognition accuracy. (Left) We show the effect of different pre-training strategies on our model. * are our reimplementations. (Right) Comparison to the state of the art on UCF101 and HMDB51. (We report the performances from the original papers.)

**Comparison of pre-training strategies.** In Table 1(Left) we compare our way of pre-training the S3D model (i.e., using CBT visual loss) to existing approaches. In particular, we consider the

Shuffle&Learn (Misra et al., 2016) and 3DRotNet (Jing et al., 2018) proxy tasks. We reimplement the two methods using S3D CNN, and pre-train them on the same Kinetics data. We also consider random initialization. We report classification results on UCF101 and HMDB51 using frozen features and fine-tuned features passed to a linear classifier. We see that our method outperforms existing training methods by a large margin.

**Comparison to existing methods.** Table 1(Right) compares the results of our method to various state-of-the art self-supervised methods. (We only report the results of fine-tuning, which are better for all methods than using frozen features.) Note that the methods differ both in architecture and training objective. First we compare against 2DCNN approaches Shuffle&Learn (Misra et al., 2016) and OPN (Lee et al., 2017). Our method outperforms both by a very large margin. This can be explained by the fact that our backbone is a 3DCNN architecture, which is much more powerful than 2D CNNs for video action recognition. Next we compare against approaches using 3DCNN architectures similar to our S3D. We also outperform all of these methods by a very large margin, and even beat the most recent approach, DPC (Han et al., 2019), by 3.8 points on UCF101 and 8.9 points on HMDB51. We believe this is due to the better contextual features that we are able to learn by using the transformer model and NCE loss.

## 4.2 LEARNING SELF-SUPERVISED TEMPORAL REPRESENTATIONS

In this section, we consider self-supervised training of representations from long videos, followed by supervised fine-tuning on various downstream tasks. To avoid running out of memory, we pre-train the S3D model on the task of classifying (short) Kinetics videos. We then freeze this feature extractor, and focus on learning longer-term temporal representations using the self-supervised CBT model. That is, we precompute short term representations $\mathbf{e}_t^x = f_{\text{enc}}(\mathbf{x}_t)$ for all videos using S3D, and focus on learning global representations $\mathbf{h}_{1:T}^x = \text{CBT}(\mathbf{e}_{1:T}^x)$.

For the self-supervised pre-training, we use unlabeled videos from the HowTo100M dataset (Miech et al., 2019b). This contains $\sim 1M$ instructional videos (details below), so the speech is informative about the vision. Therefore, we also run ASR on this dataset and use cross-modal training to compute $\mathbf{h}_{1:T}^{xy} = \text{CrossTransformer}(\mathbf{h}_{1:T}^x, \mathbf{h}_{1:T'}^y)$, where $\mathbf{h}_{1:T'}^y = \text{BERT}(y_{1:T'})$ is the output of a pretrained BERT model. To evaluate the performance of the pre-trained features (both visual and cross-modal), we consider three tasks: "action anticipation" (i.e, predicting the next action label to occur given some temporal prefix); video captioning (see section 6.1 in supplementary), and video segmentation (see section 6.2 in supplementary).

**Details on self-supervised pre-training.** We pre-train our model on HowTo100M (Miech et al., 2019b). This is a new large-scale dataset of 1.22M narrated instructional videos available on YouTube and covers among 23k different visual tasks. The average duration of a video is 6.5 minutes and there are on average 110 clips per video.

To extract visual features, we resize the videos to be 224 by 224, and compute visual features over sliding windows of 1.5 seconds (30 frames at 20 FPS) using an S3D network (Xie et al., 2018) pre-trained on the Kinetics dataset (Kay et al., 2017). We take the feature embeddings from the final `Mixed5c` block of the S3D network before the classification layer, and average pool the features spatio-temporally to get vectors of size 1024. We follow the same strategy for extracting visual features on the downstream tasks. The visual features are not fine-tuned during pre-training or when applied to downstream tasks.

To extract text features, we convert the audio track into sentences by calling the YouTube ASR API, followed by an off-the-shelf LSTM-based language model to add punctuation, thus converting the stream of words into a stream of sentences. We then follow the standard preprocessing steps from BERT (Devlin et al., 2018), and use WordPieces tokenization with the same vocabulary of 30,000 tokens. To encode ASR tokens, we take the pre-trained BERT-base architecture, which has 12 layers of Transformers, each of which has 768 hidden units and 12 attention heads.

To construct paired inputs to pre-train CBT with the cross-modal objective, we iterate over all the sentence segments in the HowTo100M dataset, and concatenate short segments until they reach the maximal length of 48 tokens. We then retrieve up to 48 visual features (72 seconds) starting at the same locations in videos. We mask out 6 out of 48 features randomly. For both the video and cross-modal transformers, we set the total hidden units per layer to 768. We fix the number of layers

to 1 for the cross-modal transformer and explore the optimal number of layers and attention heads for the video transformer. Their weights are randomly initialized.

For pre-training the CBT model on HowTo100M, we use 32 Cloud TPUs and a total batch size of 128. We use the Adam optimizer with an initial learning rate of 1e-5 and a linear decay learning rate schedule. The model is trained for 2 million iterations, which takes around 2 days.

**Details on supervised evaluation.** We evaluate the pre-trained temporal representations by transfer learning to downstream tasks. We first focus on action anticipation, whose goal is to predict the future actions by observing video sequences preceding them. In the supplementary, we also present results on video captioning and action segmentation.

For the action anticipation task, we follow the standard setup described from recent work (Abu Farha et al., 2018; Miech et al., 2019a). We consider three datasets: the Breakfast dataset (Kuehne et al., 2014) is composed of 1712 cooking videos and contains 48 fine-grained action classes; the 50Salads dataset (Stein & McKenna, 2013) contains 50 cooking videos with 17 fine-grained action classes; and the ActivityNet 200 dataset (Heilbron et al., 2015) contains 19994 YouTube videos with 200 human action classes (beyond the cooking and instructional domains). The inputs are video segments up to $T_c$ seconds before the actual action starts, and the outputs are categorical labels. For comparison with previous approaches, we set $T_c = 1$ for Breakfast and 50Salads, and $T_c = 5$ for ActivityNet.

For all experiments except for ablation on sequence lengths, we fix the input sequence length to be 72 seconds (corresponding to 48 sliding windows), features for videos shorter than 72 seconds are zero-padded. The outputs of CBT are transformed features with the same length, we take the output feature at the last non-padded position to represent the whole sequence, and put a linear classifier on top to predict the future actions. We jointly fine-tune the weights of visual transformers with the linear classifier. The text transformers and cross-modal transformers are not used during fine-tuning since only visual inputs are available. We train our model for 5 epochs using a batch size of 32 with the Adam optimizer and an initial learning rate of 1e-3. We report the top-1 accuracy on the test sets for Breakfast and 50Salads, and on the validation set for ActivityNet.

**Comparison to existing methods.** Table 2 (Left) compares to existing methods. First we compare to two existing self-supervised approaches, namely Vondrick et al. (2016) and Sun et al. (2019a). Our approach outperforms both by a very large margin. The difference with VideoBERT (Sun et al. (2019a)), which also relies on a BERT model, can be explained by the fact that it quantizes the visual features into tokens and, hence loses discriminative power. Next we compare to some recent methods that train deep classifiers end-to-end, namely (Abu Farha et al., 2018) and (Miech et al., 2019a). We outperform both by a large margin.

| Method | Self-super | Bkfst | Salads | ActNet |
|---|---|---|---|---|
| Vondrick et al. (2016) | Y | 8.1 | 6.2 | - |
| Abu Farha et al. (2018) | N | 30.1 | 30.1 | - |
| Sun et al. (2019a) | Y | 9.1 | 5.5 | - |
| Miech et al. (2019a) | N | 32.3 | - | 54.8 |
| CBT | Y | **32.7** | **39.1** | **59.8** |

| Window (in sec.) | Bkfst | | | Salads | | | ActNet | | |
|---|---|---|---|---|---|---|---|---|---|
| | AvgPool | LSTM | CBT | AvgPool | LSTM | CBT | AvgPool | LSTM | CBT |
| 1.5 | 20.3 | - | - | 30.2 | - | - | 42.9 | - | - |
| 15 | 23.5 | 24.2 | 26.3 | 36.8 | 29.7 | 36.6 | 54.2 | 54.2 | 54.4 |
| 30 | 24.2 | 25.6 | 29.8 | 34.3 | 33.9 | 37.8 | 52.9 | 57.1 | 57.5 |
| 45 | 22.7 | 26.2 | 30.8 | 32.2 | 35.3 | 38.5 | 50.7 | 57.3 | 58.8 |
| 72 | 21.9 | 27.0 | **32.7** | 26.7 | 35.6 | **39.1** | 46.2 | 57.3 | **59.8** |

Table 2: Action anticipation accuracy. (Left) Comparison to the state of the art on Breakfast, 50 Saldads, ActivityNet. Self-super = Y means the model was pre-trained in a self-supervised way, and then fine-tuned using a linear classifier. Self-super = N means the model is trained end-to-end on the specific task. (Right) Comparison with the average pooling and LSTM baselines on 50Salads Breakfast, 50Salads and ActivityNet. We vary the observation window lengths (sec.)

**Effect of video length.** In Table 2 (Right) we show the impact of the length of the training videos on the performance. We compare with two baselines, average pooling (AvgPool) and LSTM (Hochreiter & Schmidhuber, 1997). The AvgPool baseline simply computes the average of all input visual features over time. The LSTM baseline takes the same sequence of S3D features but recurrently updates its hidden states over time. The final hidden state is used for classification. We adjust the hidden unit size of LSTM to make its number of parameters comparable to CBT. We can see that CBT significantly outperforms the two baselines on all three datasets. Moreover, we can see that as the observed video length increases, the performance of CBT monotonically increases, while LSTM and AvgPool either plateaus or decreases. These results indicate that CBT is better at modeling long-term temporal context.

| Data size (%) | Cross-modal | Bkfst | Salads | ActNet |
|---|---|---|---|---|
| *0* | x | *28.8* | *35.1* | *57.4* |
| 10 | ✓ | 29.9 | 37.3 | 58.3 |
| 25 | ✓ | 30.2 | 37.3 | 58.4 |
| 50 | ✓ | 30.0 | 37.4 | 58.5 |
| 75 | ✓ | 31.4 | 38.5 | 59.3 |
| 100 | x | 29.9 | 37.6 | 57.6 |
| 100 | ✓ | **32.7** | **39.1** | **59.8** |

| L | A | Bkfst | Salads | ActNet |
|---|---|---|---|---|
| 1 | 4 | 29.6 | 34.89 | 59.3 |
| 2 | 4 | **32.7** | **39.1** | **59.8** |
| 4 | 4 | 32.7 | 38.9 | 59.2 |
| 8 | 4 | 23.2 | 6.1 | 58.0 |
| 16 | 4 | 9.17 | 5.8 | 57.5 |

| L | A | Bkfst | Salads | ActNet |
|---|---|---|---|---|
| 2 | 1 | 30.9 | 37.5 | 58.2 |
| 2 | 2 | 31.8 | 36.3 | 58.3 |
| 2 | 4 | **32.7** | 38.9 | **59.4** |
| 2 | 8 | 30.9 | **39.9** | 58.9 |
| 2 | 16 | 31.8 | 39.8 | 57.8 |

Table 3: Ablation study on the action anticipation task. We show accuracy on Breakfast, 50Salads and ActivityNet. (Left) Impact of the percentage of HowTo100M videos used, and the cross-modal objective during pre-training. 0% corresponds to no pretraining, ie. using random weights. (Middle, Right) Impact of the number of layers (L) and attention heads (A) for the visual transformers.

**Effect of dataset size and cross-modal training**. In Table 3 (Left), we study the impact of pre-training data set. As expected, pre-training with more examples leads to higher performance on all three benchmarks. We also study the impact of cross-modal training. We see this helps signficantly, especially on the smaller datasets (Breakfast and Salads).

**Effect of model size.** In Table 3 (Middle) and (Right), we study the impact of the number of layers (L) and the number of attention heads (A) for the visual transformer. Not surprisingly, model performance initially increases, but surprisingly, it then starts to decrease, in contrast to the case of NLP-BERT. We conjecture that this is because our unlabeled pre-training set is much smaller than used by the NLP-BERT model. Fortunately, our technique is quite scalable, since we can train the video representations on top of S3D features using relatively shallow transformers — our visual transformer only has 15M parameters, whereas the BERT NLP transformer has 110M parameters.

**Applications to other tasks.** In Table 4 we show the results of using of our learned temporal representation for video captioning and action segmentation. See section 6.1 and section 6.2 in the supplementary for details.

| Method | BLEU-4 | METEOR | ROUGE-L | CIDEr |
|---|---|---|---|---|
| Zhou et al. (2018c) | 4.38 | 11.55 | 27.44 | 0.38 |
| S3D | 3.24 | 9.52 | 26.09 | 0.31 |
| VideoBERT | 4.33 | 11.94 | 28.80 | 0.55 |
| CBT | **5.12** ($\pm$ 0.02) | **12.97** ($\pm$ 0.05) | **30.44** ($\pm$ 0.08) | **0.64** ($\pm$ 0.00) |

| Method | Frame Acc. |
|---|---|
| Tang et al. (2019) | 25.8 |
| Richard et al. (2018) | 21.2 |
| Ding & Xu (2018) | 34.3 |
| CBT | **53.9** |

Table 4: (Left) Video captioning results on the YouCook2 dataset (Zhou et al., 2018b). We compare with previous state-of-the-art methods by Zhou et al. (2018c) and Sun et al. (2019a), the caption decoder of all methods share the same architecture, the main difference comes from the visual encoder. (Right) Action segmentation results on the COIN dataset (Tang et al., 2019). A linear classifier is applied on the sequence of CBT output features for dense frame labeling. We compare with previous state-of-the-art methods using the standard frame accuracy metric.

# 5 CONCLUSION

We have shown how to extend the BERT model to learn representations from video in a self-supervised way, without needing vector quantization or pre-trained visual features. We have also shown how to extend this to the cross-modal setting, when ASR is available. Finally, we demonstrated that our method learns features that are far more useful than existing self-supervised methods for a variety of downstream video tasks, such as classification, captioning and segmentation. We believe that the simplicity and modularity of our method will let us scale to much larger unlabeled video datasets, which we hope will let us finally surpass supervised video pretraining (e.g., on Kinetics), just as other methods (e.g., CPC++ (Hénaff et al., 2019)) have recently surpassed supervised image pretraining (on ImageNet).

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

# 6 SUPPLEMENTARY MATERIALS

## 6.1 VIDEO CAPTIONING

In this section, we apply our model to video captioning.

**Dataset.** We pretrain our model on HowTo100M, and then use its features as input to a captioning model (details below) which is trained on the YouCook2 dataset (Zhou et al., 2018b). This contains 2000 Youtube videos of an average length of 5.26 minutes for a total of 176 hours. The annotations consist of segmentation boundaries and captions with on average 7.7 segments per video and 8.8 words per caption. We made sure that there is no overlap between the videos from our pre-training datasets and YouCook2.

**Model.** We follow the experimental setup from (Zhou et al., 2018c), where the ground truth video segmentations from YouCook2 are used to train a supervised model mapping video segments to captions. Our captioning model is a transformer with 2 layers and a hidden layer of size 128. During training we set the dropout probability to 0.4. We train our model for 10K iterations using batch size of 128 with the Adam optimizer and an initial learning rate of 1e-4. We report BLEU, METEOR and ROUGE metrics on the validation set.

**Comparison to other methods.** Table 4 shows our results. We outperform a simple baseline computed using average-pooled S3D features. We also outperform the approach of Zhou et al. (2018c) and VideoBERT Sun et al. (2019a) on all reported metrics. The comparison to VideoBERT is particularly interesting. The gains suggest that removing the quantization of video features is important for obtaining a fine-grained video representation. We also observe that the difference between CBT and VideoBERT is smaller for YouCook2 than for Breakfast and 50Salads action anticipation task, possibly because the YouCook2 dataset set is more similar to the cooking videos used for pre-training by VideoBERT.

## 6.2 ACTION SEGMENTATION

In this section, we apply our model to the task of temporal action segmentation.

**Dataset.** We pretrain our model on HowTo100M and then use its features as input to a linear classifier (details below) which is trained on the COIN dataset Tang et al. (2019). This contains 11827 instructional Youtube videos of an average length of 2.36 minutes. The annotations consist of segment boundaries and class label. On average there are 3.91 segments per video each of which lasts 14.9 seconds. There are in total 779 classes.

**Model.** We extract video features using S3D and feed the sequence to the visual transformer. We use a fixed size of 72 seconds and use zero-padding for shorter sequences. The overall clip is represented by its associated output embedding of size 768. This preprocessing step is frozen. We feed the features to a linear classifier, which we train or model for 100K iterations using batch size of 32 with the Adam optmizer and initial learning rate of 1e-3. At test time we operate on a long video with a sliding window of 72 seconds.

**Comparison to existing approaches.** In Table 4 we compare CBT against various state of the art approaches using the frame acuracy as metric, including (Tang et al., 2019), (Richard et al., 2018) and (Ding & Xu, 2018). We outperform them by a large margin state (+19.6 points).

