# OpenReview forum: "Learning Video Representations using Contrastive Bidirectional Transformer"
_ICLR.cc/2020/Conference — Reject_

### Official Review · AnonReviewer3 · 2019-10-22
**Official Blind Review #3**

**Rating:** 6

**Review:**

This paper presents a novel method to extract cross-modal text-visual embeddings on the HowTo 100M corpus. The core idea is to extend previous work on clip-level embeddings (e.g. the max-margin ranking loss proposed for HowTo 100M) to a transformer architecture which takes into account the entire context of a video, which should lead to better learned representations and improved performance in downstream tasks. In addition, the max-margin loss is replaced by noise contrastive estimation.

The paper is well written and explains the main problem well, however I do have a few questions:
- I do not understand the sentence "However, for images and videos, the inputs are real-valued vectors." (Section 3.2) - Transformers are being used for speech recognition or speech translation - the input features are not the problem. The outputs are assumed to be discrete (in the original formulation)
- Why not directly compare your approach to the approach presented in (Miech, 2019c) - it would be interesting to see a direct comparison, but as far as I can tell, there is no overlap in tasks?
- What is the influence of adding punctuation to the ASR output, how good is it, and how good is the underlying ASR? Why did you not use the original text annotations provided by HowTo 100M, but run the audio through Google ASR (again?) It would be good to know how good the ASR is, and if adding in punctuation post-hoc works well, and how this influences your use with a pre-trained BERT model. My guess is that the BERT model will be happy as long as it sees a "." at the end?
- Also, would it be possible to compare the results of your work with some of the work in (Miech, 2019c) - it almost seems that your work avoids comparing your results to this previous work.


**Experience Assessment:**

I have published one or two papers in this area.

**Review Assessment: Checking Correctness Of Derivations And Theory:**

I assessed the sensibility of the derivations and theory.

**Review Assessment: Checking Correctness Of Experiments:**

I assessed the sensibility of the experiments.

**Review Assessment: Thoroughness In Paper Reading:**

I read the paper at least twice and used my best judgement in assessing the paper.

---

> ### Author Response · Authors · 2019-11-14
> **Our response**
>
> Thank you for your positive feedback.
>
> Comparison to HowTo100M:
>
> Although we use the HowTo100M dataset for pre-training, there are key differences to (Miech, 2019c):
> 1. Miech et al. improve text-video embedding by training on HowTo100M and show the gain by transferring it to the text to video retrieval task. In comparison, CBT focuses on learning generic visual and temporal features, with or without using text-video correspondences. We show that CBT can be transferred to various downstream tasks, such as classification, anticipation, segmentation and captioning.
> 2. Miech et al. assume the visual features to be pre-trained and fixed, while CBT can be applied for self-supervised visual representation learning, as shown in Table 1.
>
> Direct comparison to (Miech2019c):
> We evaluate our cross-modal model pre-trained on HowTo100M with the same preprocessing as in (Miech2019c), i.e. with short clips. We evaluate on the MSR-VTT clip retrieval benchmark (zero-shot settings) and can observe that we outperform their approach, see below. We will add this results to the final version of the paper if accepted.
> HowTo100M (table 6):    R@1: 7.5   R@5: 21.2   R@10: 29.6   median R: 38
> Ours:                                 R@1: 8.3   R@5: 23.3   R@10: 33.2   median R: 30
>
> “The inputs are real-valued vectors”:
> Thank you for pointing this out, we will clarify in the final version.
>
> Influence of ASR and punctuation:
> The video clips and speech released by HowTo100M were preprocessed and broken into short segments. To learn long-term temporal features with the transformer, our approach requires longer input clips. Hence, we re-extract the ASR with the same algorithm as HowTo100M and run punctuation to get longer semantic coherent text segments (sentences). Furthermore, we concatenate several consecutive sentences to obtain even longer sequences of video-ASR training data.

---

### Official Review · AnonReviewer2 · 2019-10-23
**Official Blind Review #2**

**Rating:** 6

**Review:**

This is one of those papers where the number of experiments conducted to produce the results is beyond the capabilities of "almost all" research groups. From the paper: "we use 32 Cloud TPUs. The model is trained for 2 million iterations, which takes around 2 days." However, with that being said, it's a good paper of general interest to the community.

The paper focuses on self-supervised learning in video, and combines two contributions. The first is using a noise contrastive estimation loss (2016) which can be used for any visual dataset. The second is a cross-modal (BERT) model that requires language and vision. A few modifications over other BERT flavours are introduced. The cross-modal BERT is not tested alone, however when added to the NCE loss function, seems to suit a range of downstream tasks from classification to anticipation and captioning. NCE alone seems to clearly produce better results over published results, however these are not compared like-to-like, as published results are used for this comparison.

The paper is full of technical details to reproduce the results. This makes the main novelty is actually in showing that this approach works. However, the approach is technically sound and up to my knowledge has not been attempted before.

**Experience Assessment:**

I have published in this field for several years.

**Review Assessment: Checking Correctness Of Derivations And Theory:**

I assessed the sensibility of the derivations and theory.

**Review Assessment: Checking Correctness Of Experiments:**

I assessed the sensibility of the experiments.

**Review Assessment: Thoroughness In Paper Reading:**

I read the paper at least twice and used my best judgement in assessing the paper.

---

> ### Author Response · Authors · 2019-11-14
> **Our response**
>
> Thank you for your positive feedback.

---

### Official Review · AnonReviewer4 · 2019-11-04
**Official Blind Review #4**

**Rating:** 6

**Review:**

This paper is about a self-supervised video representation with a multi-modal learning process that the authors then use for performance on a variety of tasks. The main contribution of the paper is a successful effort to incorporate BERT-like models into vision tasks. As is detailed in the related work, the field has been inching towards this but without as much success as this paper has.

My main criticism of the paper is that it feels like there is everything and a bag of chips happening; It's exceptionally hard to tease apart what is the main contribution to its success. I mostly came away from the paper thinking that it was good to see an existence proof of successfully incorporating the result, but not having really understood anything more wrt why or how this works. Other than it being a good idea to have a bigger model and more varied types of gradients, it's unclear what this model does that distinguishes it from other approaches.

On a more specific critique level, why use COIN? And why compare on a frame accuracy metric? The comparison to Ding & Xu seems a bit odd given that they don't assume access to annotations but rather to video transcripts. There are other datasets that you could make use of here that are more applicable, like Thumos14 or ActivityNet. I understand that this is a small section, but arguably the paper would be stronger if more time was spent on the main result than on this sidebar.

Overall, I'm giving it a weak accept because I do think that the community should be aware of this paper's result.

**Experience Assessment:**

I have read many papers in this area.

**Review Assessment: Checking Correctness Of Derivations And Theory:**

N/A

**Review Assessment: Checking Correctness Of Experiments:**

I assessed the sensibility of the experiments.

**Review Assessment: Thoroughness In Paper Reading:**

I read the paper at least twice and used my best judgement in assessing the paper.

---

> ### Author Response · Authors · 2019-11-14
> **Our response**
>
> Thank you for your positive feedback!
>
> In the following we summarize our main contributions:
> 1. CBT objective for self-supervised visual representation learning. We study the impact of the CBT objective, which uses long temporal information as self-supervision, in Table 1. We observe that CBT outperforms alternative training objectives based on local spatial (3DRotNet) and temporal (Shuffle&Learn) objectives significantly, when they were pre-trained on the same Kinetics data with the same backbone ConvNet (Table 1 left). Our CBT method also outperforms the state of the art (Table 1 right).
>
> 2. CBT and cross-modal objectives for temporal representation learning. In Table 2 (left) and Table 4 (left) we compare our method with VideoBERT, which applies vector quantization on visual features. We found CBT outperforms VideoBERT significantly. In Table 3 (left), we observe that the cross-modal objective further improves the performance on action anticipation tasks.
>
> Choice of datasets:
>
> The action segmentation accuracy metric, along with the baselines we compare with, were quoted from Table 3 of the COIN dataset paper. We note that their fully-supervised baseline (Tang et al. 2019) performed worse than the weakly-supervised baseline (Ding & Xu 2018). We will make the supervision type clear in the final version.
> Besides the COIN dataset, we do provide ActivityNet evaluations in Table 2&3, along with other benchmarks such as HMDB (Table 1), UCF (Table 1), Breakfast (Table 2&3) and YouCook (Table 4). This range of datasets helps us understand the performance of CBT in diverse visual domains.

---

### Decision · Program_Chairs · 2019-12-19

**Decision:**

Reject

**Comment:**

This paper studies self-supervised video representations with a multi-modal learning process that the authors then use for performance on a variety of tasks. The main contribution of the paper is a successful effort to incorporate BERT-like models into vision tasks.

Reviewers acknowledged the extensive empirical evaluation and the good performance of the approach. However, they raised some concerns about the lack of clarity and the absence of analysis and interpretation of the results. The AC shares this view, and recommends rejection at this time, encouraging the authors to revise their work addressing these analysis and clarity questions.